# Optimization of Photodynamic Therapy in Dermatology: The Role of Light Fractionation

**DOI:** 10.3390/ijms26168054

**Published:** 2025-08-20

**Authors:** Luis Alonso-Mtz de Salinas, Emilio Garcia-Mouronte, Jorge Naharro-Rodriguez, Luis Alfonso Perez-Gonzalez, Montserrat Fernández-Guarino

**Affiliations:** Dermatology Department, Hospital Universitario Ramon y Cajal, Carretera M-607 km 9.1, 28034 Madrid, Spain; emilio.garcia.mouronte@gmail.com (E.G.-M.); jorgenrmed@gmail.com (J.N.-R.); pg.l.alfonso@gmail.com (L.A.P.-G.)

**Keywords:** photodynamic therapy, light fractionation, actinic keratosis, basal cell carcinoma, 5-aminolevulinic acid, methyl aminolevulinate, reactive oxygen species, non-melanoma skin cancer

## Abstract

Photodynamic therapy (PDT) has become a widely used modality for treating actinic keratosis (AK) and non-melanoma skin cancers (NMSC), as well as other inflammatory or infectious diseases. Despite its efficacy, limitations such as incomplete responses and pain have motivated the exploration of protocol enhancements. This review examines the clinical and biological rationale for light fractionation—dividing the total light dose into two separate exposures with a dark interval—as a strategy to improve PDT outcomes. We reviewed preclinical and clinical studies evaluating fractionated illumination using 5-aminolevulinic acid (ALA) or methyl aminolevulinate (MAL). The findings consistently demonstrate superior efficacy of fractionated schemes, particularly with ALA, showing higher complete response rates in AK, superficial basal cell carcinoma (sBCC), and Bowen’s disease (BD), and improved long-term tumor control compared to single illumination. The better outcomes are attributed to increased reactive oxygen species (ROS) generation following tissue reoxygenation during the dark interval and greater susceptibility of partially damaged cells to subsequent illumination. Fractionated PDT also shows a favorable safety and cosmetic profile. These results support considering light fractionation protocols as a standard approach for optimizing PDT efficacy in dermatologic oncology, particularly in lesions with limited depth and high recurrence risk.

## 1. Introduction

Photodynamic therapy (PDT) is a therapeutic modality that combines a photosensitizer (PS), visible light, and molecular oxygen to induce selective destruction of abnormal tissue. It has been widely employed in dermatology, particularly for the treatment of actinic keratosis (AK), non-melanoma skin cancers, acne, photoaging, microbial infections, among others [1,2,3]. Its mechanism of action primarily involves the generation of singlet oxygen (^1^O_2_) and reactive oxygen species (ROS), which trigger apoptosis, necrosis, and immune modulation in target cells [4,5].

The efficacy of PDT depends on multiple factors, including the type and concentration of the PS, incubation time, and the characteristics of the light source, such as wavelength, power density, and fluence. Among these parameters, light fractionation—dividing the total irradiation time into multiple segments with rest intervals—has emerged as a promising strategy to enhance therapeutic outcomes [6].

## 2. Fundamentals of PDT

PDT is based on the interaction between three essential components: a photosensitizing agent, a specific light source, and molecular oxygen. When exposed to light of an appropriate wavelength, the PS, which selectively accumulates in tumor cells, is activated and undergoes a series of photochemical reactions that generate ROS, including ^1^O_2_. After light absorption, the PS transitions from its ground singlet state to an excited singlet state and then to a long-lived triplet state. In this state, it can either transfer energy directly to molecular oxygen to form ^1^O_2_ (Type II mechanism) or engage in electron transfer reactions to produce other ROS such as superoxide anion, hydrogen peroxide, and hydroxyl radicals (Type I mechanism). These ROS damage cellular structures, leading to tumor destruction through both direct cytotoxicity and indirect effects. Although both pathways contribute to cytotoxicity, the Type II reaction is generally considered the predominant and most potent mechanism, as ^1^O_2_ is highly reactive and has the greatest impact on cellular damage. Protoporphyrin IX (PpIX), the active PS from aminolevulinic acid, localizes mainly to the inner mitochondrial membrane. Upon illumination, ROS oxidize mitochondrial components, triggering cytochrome c release and activation of the apoptotic pathway and, in cases of severe damage, causing cell death by necrosis [7,8,9].

The most commonly used PS in dermatology are 5-aminolevulinic acid (ALA) and methyl aminolevulinate (MAL), both of which are prodrugs converted intracellularly into PpIX. PpIX accumulates preferentially in abnormal or rapidly proliferating cells and is highly sensitive to light activation [5,10].

Light sources used in PDT vary and include light-emitting diodes (LEDs), lasers, fluorescent lamps, and even natural daylight. The wavelength of light determines its penetration depth, with red light (625–700 nm) reaching deeper skin layers compared to blue light (400–500 nm) [8,11]. Figure 1 schematizes the mechanism of action of PDT.

Key parameters that influence PDT outcomes include the following:−PS factors: type, concentration, incubation time, and temperature.−Light parameters: wavelength, fluence, irradiance, and exposure duration.−Tissue characteristics: oxygen availability, pigmentation, and vascularization.

Understanding and optimizing these variables is critical for maximizing therapeutic efficacy while minimizing adverse effects, such as erythema, edema, and pain. These fundamental principles form the basis for exploring advanced strategies like light fractionation to further refine PDT protocols.

## 3. Light Fractionation Concept

Light fractionation is defined as the delivery of the total therapeutic light dose in two or more discrete fractions separated by dark intervals, rather than by a single continuous exposure. This strategy, initially inspired by radiotherapy protocols, is based on the oxygen-dependent nature of PDT [12]. During continuous illumination, rapid oxygen consumption can lead to hypoxia, thus limiting the effectiveness of singlet oxygen-mediated cytotoxicity. Introducing a dark interval allows for the partial restoration of oxygen levels within the treated tissue, leading to more effective ROS generation upon subsequent light exposure (Figure 2) [12,13,14].

A typical two-step fractionated illumination protocol involves an initial sublethal light dose, followed by a rest period, and a second, higher-dose exposure. This scheme has been shown to induce greater phototoxic effects, enhanced apoptosis, and superior tumor control compared to single-dose illumination [13,15,16,17]. Furthermore, evidence from both preclinical and clinical studies demonstrates that fractionated illumination can result in higher complete response rates, particularly in superficial basal cell carcinoma (sBCC) and AK [6,12,18].

In summary, light fractionation represents a significant advancement in PDT methodology, offering improved outcomes through optimized ROS production, reduced pain perception, and better tissue selectivity.

## 4. Mechanistic Insights from Preclinical Models of Light Fractionation in PDT

Most of the current understanding of the effects of light fractionation in PDT comes from preclinical research. Experimental work has elucidated the biological pathways through which it may act. These models allow detailed exploration of processes such as oxygen dynamics, PS distribution, and vascular effects, that cannot be easily studied in clinical settings. Below, we integrate the main mechanistic hypotheses with the preclinical evidence that supports them.

### 4.1. ROS Dynamics and Tissue Reoxygenation

A fundamental limitation of continuous PDT is rapid oxygen depletion during illumination, because ROS generation, particularly ^1^O_2_, is highly oxygen dependent. During the initial illumination phase, photochemical reactions consume O_2_ faster than the vasculature can resupply it, causing local hypoxia and reducing cytotoxic efficacy. Fractionating light delivery with a dark interval allows tissue reoxygenation, restoring oxygen tension, and enabling a second wave of ROS production during the subsequent illumination [6,9,17,19,20].

Preclinical Photofrin-PDT studies using reactive oxygen species explicit dosimetry (ROSED) have shown that, although total reacted ROS values may be similar between continuous and fractionated regimens, long-term tumor control is markedly higher with fractionation. This improvement coincides with measurable recovery of both tumor oxygenation and intratumoral Photofrin concentration during the dark interval, maintaining higher average substrate levels throughout treatment. The enhanced efficacy is therefore attributed not only to restored oxygen availability but also to the generation of ROS in a metabolic and microenvironmental context primed by sublethal photodynamic injury from the first light fraction. Therefore, the timing and physiological conditions during ROS generation may be as critical as their absolute quantity for achieving optimal PDT outcomes [21,22,23].

In line with these findings, experiments in mouse skin treated with BF-200 ALA have demonstrated that fractionated light (20 + 80 J/cm^2^, 2 h interval) produced greater visual skin damage than single-dose illumination. Fluorescence imaging confirmed deeper dermal accumulation of PpIX under fractionated protocols [6].

### 4.2. Sublethal Damage and PpIX Dynamics

The biological rationale behind this approach lies in the synergistic effect of sublethal photodynamic damage during the first light dose, initiating oxidative stress and partial apoptosis, followed by a dark interval that allows tissue reoxygenation. The subsequent light exposure amplifies the cytotoxic response. This enhanced efficacy appears to be PS-dependent, as studies have shown that PpIX accumulation in endothelial cells and tumor stromal components is higher following ALA application than with esterified derivatives like MAL [6,24].

After ALA incubation, the first illumination primarily affects the mitochondria, where PpIX accumulates and generates ^1^O_2_, causing partial membrane permeabilization, mitochondrial depolarization, oxidative injury, and partial release of cytochrome c into the cytosol [6,25,26]. Because ^1^O_2_ has a very short diffusion range, PpIX must be closely associated with specific molecular targets to efficiently trigger cytochrome c release. Such targets include the peripheral benzodiazepine receptor (TSPO) within the mitochondrial permeability-transition pore complex and cardiolipin, a phospholipid anchoring cytochrome c. These mitochondrial events initiate pro-apoptotic signaling without completing cell death, leading to a transiently sensitized state [26].

In vitro studies confirm that cells exposed to a sublethal first light fraction show significantly reduced survival after a second illumination, despite similar PpIX levels, supporting a mechanism beyond PS re-accumulation [26]. At the molecular level, this sensitization is associated with the early activation of pro-apoptotic pathways, including BAX translocation, mitochondrial outer membrane permeabilization, and partial caspase-3 activation. Simultaneously, anti-apoptotic regulators like BCL-2 may be downregulated, further tipping the balance toward cell death upon re-illumination [25]. It should be emphasized that this priming effect is achieved only when the first light fraction delivers a non-excessively energetic dose. Excessive initial illumination can overwhelm the early apoptotic priming mechanisms, rapidly committing cells to necrosis during the first phase and thereby abolishing the effect of light fractionation [27].

During the dark interval between light exposures, cells that have been only partially damaged by the initial illumination retain their ability to synthesize heme and continue producing PpIX. The amount of PpIX generated during this period depends on the light dose delivered in the first exposure; lower initial doses generally result in a higher final concentration of PpIX [28].

The subsequent, more energetic second light fraction then acts on a cellular environment already destabilized by oxidative stress, leading to irreversible apoptosis or secondary necrosis. These sequential events occurring in tumor cells are schematically summarized in Figure 3.

In line with these mechanistic findings, advanced 3D in vitro tumor models have shown a more pronounced phototoxic effect with fractionated light delivery compared to a single continuous exposure of equal total dose. In melanoma spheroids that replicate the architecture and microenvironment of solid tumors, this stronger response was linked to the cumulative sublethal injury from the first light fraction, which may facilitate deeper PS penetration and redistribution before the second illumination [29].

### 4.3. Vascular Disruption

PDT also targets the tumor vasculature. Light fractionation has been shown to intensify endothelial cell damage, leading to enhanced vascular shutdown. Immunofluorescence studies in mouse models revealed a pronounced loss of CD144 (an integral component of endothelial adherens junction complex) and CD31 expression, indicating greater endothelial detachment and vessel collapse. Intravital microscopy data further suggest a trend toward stronger arteriole constriction after BF-200 ALA-PDT delivered with light fractionation compared to single illumination, although this difference did not reach statistical significance. This effect contributes both to direct tumor ischemia and to limiting metastatic potential by compromising the tumor microcirculation [6,30].

Additional preclinical data support these findings from a functional perspective. In a murine squamous cell carcinoma model, fractionated ALA-PDT induced greater acute vascular damage than single-illumination PDT, as evidenced by increased edema formation and a higher incidence of scab development in the treated area, despite no measurable differences in blood volume fraction. Optical spectroscopy confirmed that fractionated illumination allowed recovery of microvascular oxygen saturation during the dark interval, suggesting enhanced perfusion and oxygen availability before the second light fraction [20]. In a rat skinfold tumor model, a two-fraction protocol led to full vascular stasis and sustained tumor necrosis, without widespread damage to surrounding tissue, an outcome not achieved with a single dose [31].

## 5. Clinical Evidence for Light Fractionation in PDT

The clinical application of light fractionation in PDT has garnered increasing attention due to its potential to improve therapeutic efficacy and patient tolerance. A number of randomized controlled trials (RCT) and prospective clinical studies have evaluated this approach across a range of dermatological conditions, with AK and basal cell carcinoma (BCC) being the most frequently investigated.

In a randomized intraindividual trial, Sotiriou et al. focused on AK lesions of the face and scalp, comparing conventional ALA-PDT (two sessions of 75 J/cm^2^, 7 days apart) with a single-session fractionated protocol (20 J/cm^2^ + 2 h dark interval + 80 J/cm^2^). The fractionated scheme achieved significantly higher complete response rates at both 3 and 12 months, particularly in grade I and II lesions [16].

A retrospective multicenter study evaluated the efficacy of the same fractionated protocol in a mixed population of 552 lesions, including 70 AKs, 32 Bowen’s disease (BD), 430 superficial BCC (sBCC) and 20 nodular BCC (nBCC). The reported complete response rate was 98% for AK, 84% for BD, 97% for sBCC, and 80% for nBCC, after 2 years of follow-up. Cosmetic outcomes were rated as good or excellent in the vast majority of cases [15].

In another RCT, Puizina-Ivić et al. compared single (100 J/cm^2^) vs. fractionated illumination (50 + 50 J/cm^2^ with 2 h interval) after topical 5-ALA in 36 AKs and 15 BD lesions. Histological evaluation at 24 weeks showed that residual tumor was present in 75% of AK and 66.6% of BD lesions in the single-illumination group, compared to only 12.5% of AK and 22.2% of BD in the fractionated group. This difference was statistically significant, confirming a superior therapeutic effect of the fractionated scheme in both lesion types [17].

Additionally, in a RCT, de Haas et al. compared a single illumination (75 J/cm^2^) with a fractionated scheme of 20 + 80 J/cm^2^ (2 h dark interval) in 505 sBCC lesions. At 12 months, the complete response rate was significantly higher with fractionation (97% vs. 89%, *p* = 0.002), confirming the clinical superiority of the split-dose approach. Notably, even when limited to histologically confirmed sBCC, the benefit persisted (98% vs. 85%, *p* = 0.0003) [13]. A subsequent 5-year follow-up confirmed superior long-term efficacy, with 88% complete response (CR) vs. 75% for single illumination (*p* = 0.0002), supporting the durability of the fractionated approach [32].

Finally, Kessels et al. compared fractionated ALA-PDT (20% ALA, 20 + 80 J/cm^2^) with a two-stage MAL-PDT protocol in patients with sBCC. While the difference in clearance at 12 months did not reach statistical significance (92.3% vs. 83.4%, *p* = 0.091), the ALA-based fractionated regimen showed a trend toward better efficacy [18]. However, in the extended 5-year follow-up, this initial advantage was not maintained. The long-term tumor-free survival was significantly higher for the conventional MAL-PDT regimen compared to fractionated ALA-PDT (76.5% vs. 70.7%), with most late recurrences occurring in the ALA-treated group [33].

In nodular BCC (nBCC), which is typically more treatment-resistant, Roozeboom et al. conducted a RCT comparing fractionated ALA-PDT after partial debulking to surgical excision. Although surgery remained superior in terms of 5-year recurrence (2.3% vs. 30.7%), PDT was effective in thinner tumors (≤0.7 mm), achieving a 94.4% recurrence-free survival in that subgroup [34,35].

A randomized study by de Haas et al. compared a standard single illumination (75 J/cm^2^) with a two-step protocol (20 J/cm^2^ + 2 h dark interval + 80 J/cm^2^) in 50 BD lesions. While the difference in response rates at 12 months (88% vs. 80%) did not reach statistical significance, the authors noted a trend toward improved efficacy and cosmetic outcomes with fractionation protocol [36].

The technique has also been tested beyond traditional oncologic indications. In actinic cheilitis, Suárez-Pérez et al. reported excellent clinical and cosmetic outcomes after a single fractionated session of MAL-PDT, with sustained lesion clearance up to 18 months, although no control group was included [37].

In psoriasis, Smits et al. reported clinical and immunohistochemical improvement using fractionated ALA-PDT in stable plaque lesions. While results showed reduced proliferation and T-cell infiltration, the therapeutic impact was variable, and the risk of Koebner phenomenon raises concerns about its broader applicability, particularly in active or unstable disease [38].

A novel application comes from Khan et al., who evaluated fractionated MAL-PDT in cutaneous leishmaniasis. Their protocol delivered three light doses (initial, 2 h, and 16 h intervals) and showed a 91.4% complete response rate vs. 76% in the single-illumination group. Patients in the fractionated arm also reported significantly less pain and discomfort during treatment [39].

A complete overview of these clinical studies—including their methodologies, patient populations, PS, protocols, and main outcomes—is provided in Table 1.

Overall, these clinical studies consistently demonstrate that light fractionation enhances treatment response, particularly in superficial lesions such as AK and sBCC. Benefits include higher lesion clearance rates, reduced recurrence, potential for single-session therapy and improved patient comfort and tolerability.

The clinical outcomes are influenced by factors such as the type of PS (ALA vs. MAL), depth and nature of the lesion, and fractionation timing. While more head-to-head comparisons are warranted, especially in deeper tumors and non-oncologic conditions, the current evidence supports the integration of fractionated light protocols as a refined, evidence-based enhancement to conventional PDT.

Several alternative PDT protocols have been proposed that deviate from the classical light fractionation model, which delivers two large light fractions separated by a long dark interval without PS re-application. The Flexitheralight approach delivers repeated short pulses of low irradiance red light separated by brief dark intervals (1 min light/2 min dark) within the same session after a short MAL incubation, allowing continuous micro-reoxygenation and significantly reducing pain, while maintaining efficacy in AK [40]. Moreover, two protocols with a single-visit PDT for BCC have been proposed as alternatives to classical light fractionation. Both deliver two illuminations on the same day separated by a dark interval (~90 min) with MAL re-application before the second light. In a multi-arm trial, the most effective schedule combined re-application, a 90 min dark interval, and higher second-fraction fluence, achieving a 95.4% complete response at 30 days [41]. In a long-term study for nBCC, the same-day approach yielded higher 30-day clearance (93.3% vs. 85%) and better 5-year recurrence-free survival (80.6% vs. 69.0%) than the standard one-week-apart regimen [42]. However, these designs do not meet the definition of classical light fractionation due to PS re-application.

## 6. Influencing Factors, Limitations, Clinical Implications, and Future Directions of Light Fractionation in PDT

The therapeutic success of light fractionation in PDT is determined by a multifactorial interplay between biological variables, treatment parameters, and disease-specific contexts. While the concept of fractionated light delivery offers significant advantages, its optimization and widespread clinical implementation require a nuanced understanding of influencing factors, practical limitations, and future research needs.

### 6.1. Influencing Factors

Several parameters critically modulate the efficacy of light fractionation:−PS type and pharmacokinetics: ALA-based protocols tend to benefit more from light fractionation than MAL-based ones, largely due to differences in PpIX accumulation, subcellular localization, and vascular targeting. ALA leads to broader and deeper PpIX distribution, especially in endothelial and perivascular zones, which enhances the cytotoxic effect of the second illumination.−Tissue oxygenation and perfusion: The therapeutic window between light fractions must be adequate to allow for tissue reoxygenation. This is influenced by lesion vascularity, anatomical site, and underlying patient conditions. Poorly perfused or hypoxic tissues may fail to benefit fully from fractionation unless additional oxygenation strategies are employed.−Tumor or lesion characteristics: Superficial lesions such as grade I/II AK or sBCC respond more favorably to fractionated PDT. In contrast, thicker or nodular tumors, while potentially responsive, may require debulking or enhanced delivery strategies to achieve comparable efficacy.−Protocol design: Fractionation schedules typically involve a low initial fluence, a 2 h dark interval, and a higher second fluence. However, variations exist, including multifractionated schemes or personalized intervals based on real-time oxygen monitoring, which remain under investigation.

### 6.2. Limitations

Clinically, light fractionation offers key advantages. By improving tissue oxygenation and enhancing PS activation during the second illumination, it increases oxidative damage selectively in tumor tissue. This translates into higher response rates, lower recurrence, and, in some cases, effective single-session treatment. Moreover, some studies report better tolerability and reduced pain, likely due to decreased acute inflammation.

However, these benefits must be weighed against a series of relevant limitations. First, the technique inherently extends treatment duration, posing logistical challenges in routine practice and requiring specific equipment such as programmable light sources and appropriate patient immobilization setups. Second, while the evidence supporting light fractionation is promising, experimental data indicate that its benefits may not extend to all tumor oxygenation states. In well oxygenated lesions, dark intervals between light fractions can allow glutathione reductase to regenerate reduced glutathione (GSH), enhancing the cell’s ability to neutralize PDT-generated ROS and thereby lowering cytotoxicity compared with continuous illumination. This effect has been demonstrated in vitro and could be reversed by pharmacological inhibition of glutathione reductase, suggesting that in well perfused, non-hypoxic lesions, fractionation could be less effective unless antioxidant defenses are addressed [43]. These molecular considerations further underscore the fact that the clinical data remain limited, particularly for AK—a principal indication for topical PDT. Most studies addressing AK include small cohorts or mixed lesion types, complicating the extrapolation of results.

Another major issue is the heterogeneity of protocols across studies. There is substantial variability in PS, fluence values, fractionation intervals, and light doses, making direct comparisons difficult and hindering consensus on optimal parameters. In fact, some of the superior outcomes attributed to light fractionation may partly derive from higher cumulative light doses, rather than the temporal modulation per se.

Lastly, despite favorable findings, the adoption of light fractionation in real-world practice remains limited. There is a lack of recent, large-scale studies, and its use has not been widely standardized or integrated into clinical guidelines. This gap between clinical evidence and practice reflects the need for further research, particularly head-to-head trials in high-prevalence conditions like AK, to consolidate its role within dermatologic PDT protocols.

### 6.3. Clinical Implications and Future Directions

The growing clinical evidence positions light fractionation as a valuable enhancement to standard PDT, particularly in superficial dermatologic oncology. Its integration into routine care requires harmonization of protocols across studies, allowing for robust comparative analysis and clearer guidelines. Establishing patient selection criteria based on lesion characteristics—such as depth, vascularity, and anatomical location—will be essential to optimize therapeutic outcomes. Technological developments, including the availability of programmable light systems and dosimetric feedback tools, will also play a pivotal role in expanding its clinical feasibility.

Looking ahead, future research should aim to explore molecular profiling, which could help identify biomarkers predictive of an enhanced response to fractionated protocols and facilitating personalized treatment planning. Moreover, the development of real-time feedback systems to monitor oxygenation status and PpIX dynamics could open the door to adaptive illumination protocols, further refining efficacy and safety.

Beyond optimizing illumination strategies, innovation in PS design may further expand the potential of fractionated PDT. Emerging studies have described bifunctional PS that not only generate ^1^O_2_ during light exposure but also store it chemically in the form of reversible endoperoxides, enabling sustained release of cytotoxic ROS during dark intervals. By maintaining ROS generation throughout the light–dark cycle, this approach could mitigate the impact of PDT-induced hypoxia and reduce dependence on tissue reoxygenation between fractions. Although still in experimental stages, such agents represent a promising direction for improving outcomes in hypoxic or poorly vascularized tumors [44,45].

An emerging approach to further enhance the therapeutic potential of fractionated PDT involves the use of persistent luminescent nanoparticles (PLNPs) as internal light sources. These particles can be “charged” with a brief pulse of external light and then continue to emit low level luminescence for several minutes, even in complete darkness. When coupled to a conventional PS, this afterglow can sustain ^1^O_2_ production throughout the light–dark cycle, effectively extending the therapeutic action of each fraction. In preclinical in vitro models, PLNP-PS conjugates have shown substantially greater cytotoxicity under fractionated protocols than with continuous illumination at the same total light dose, without introducing additional toxicity from the nanoparticles themselves. This strategy could further improve the efficiency of fractionated PDT, particularly in scenarios where light delivery is technically challenging [46]. Additionally, iron chelators like desferrioxamine (DFO) were explored to increase intracellular PpIX levels by inhibiting ferrochelatase. However, the combination of DFO with fractionated PDT did not produce a clear synergistic effect on long-term tumor control [20]. More recently, nanocarrier strategies co-delivering 5-ALA with DFO have demonstrated increased PpIX accumulation and enhanced ALA-PDT efficacy by simultaneously limiting heme synthesis and impairing DNA repair [47].

Complementary strategies are being developed to optimize PS delivery, activity, and tissue penetration, thereby broadening the therapeutic scope of PDT [48]. Lasers, such as fractional ablative lasers (AFL) or fractional CO_2_ lasers, serve as a potent pre-treatment to significantly improve the penetration of topical PS into the target tissue [49]. PDT also benefits from combination with specific chemical agents designed to boost its effectiveness [50]. The topical immunomodulator imiquimod has also shown promise when combined with PDT, suggesting improved clearance rates and reduced recurrences in sBCC [51]. Furthermore, physical strategies enhance PDT delivery and patient comfort; these include curettage for partial debulking of thick lesions prior to treatment, and microneedling to improve PS uptake [52]. Recent experimental research investigates novel approaches such as inducing ferroptosis, a regulated form of cell death driven by iron-dependent lipid peroxidation, which can overwhelm the tumor’s antioxidant defenses and complement apoptosis-based killing. Other strategies include inhibiting key antioxidant systems such as glutathione, and using nanocarriers to co-deliver PS with pro-oxidant or pro-apoptotic agents [53,54,55]. Topical nanocarriers, such as polymeric nanoparticles, liposomes, and dendritic systems, have demonstrated enhanced dermal penetration, controlled PS release, notably increasing phototoxic efficacy while reducing off-target toxicity [56]. Nevertheless, these approaches have not yet been specifically tested in the context of fractionated PDT, but they are likely to be an important focus for future research in this field.

## 7. Discussion

Light fractionation represents a significant advancement in the optimization of PDT, particularly in dermatological oncology. By dividing light delivery into two or more temporally separated exposures, this technique enhances the biological efficacy of PDT through mechanisms such as improved tissue oxygenation, sublethal cellular sensitization, increased ROS production, and amplified vascular disruption. Both preclinical models and clinical trials show that fractionated illumination results in higher clearance rates, prolonged disease-free intervals, and, in some cases, better cosmetic and pain outcomes compared to conventional single-dose illumination. The strong agreement between these findings in superficial lesions underscores the relevance of mechanistic data. Processes such as tissue reoxygenation, PpIX re-accumulation and increased vascular injury offer a coherent biological rationale for the superior outcomes observed in real-world series. However, the way in which these mechanisms vary across lesion types and cellular contexts, remain poorly understood. In particular, the contributions of immune modulation, vascular remodeling, and oxidative stress dynamics have yet to be clarified. Addressing these gaps could lead to the identification of predictive biomarkers and the development of synergistic adjuvants, ultimately enabling more effective fractionated PDT protocols.

Clinical evidence is stronger for superficial lesions such as sBCC or AK, where light fractionation has demonstrated therapeutic benefit. The use of ALA as a PS appears to be more favorable to these benefits than MAL, possibly due to differences in pharmacokinetics and tissue penetration. As a technique, light fractionation is safe, biologically well-founded, and increasingly supported by real-world data. Moreover, the mechanistic rationale outlined above suggests potential for combination with complementary strategies already in dermatologic use, such as fractional ablative lasers or microneedling, to enhance PS penetration, as well as with pharmacological approaches that modulate oxidative stress. Although these combinations have been explored mainly in conventional PDT, their integration with light-fractionated regimens could potentiate the synergistic effects of light fractionation and deserves further investigation. Beyond dermatologic oncology, new molecular strategies such as tumor-activatable PS and oxygen-independent PDT approaches are under development and may complement light-fractionated regimens by offering additional modes of action [57].

Despite its promise, the integration of light fractionation into routine PDT practice raises important considerations. One of the central challenges is the lack of standardization across protocols. While the 20 J/cm^2^ + 2 h + 80 J/cm^2^ scheme using ALA as the PS has been widely adopted, alternative fractionation designs (e.g., multifractionated regimens or patient-adapted intervals) exist, complicating comparative analysis. Without harmonized parameters and reporting standards, it remains challenging to define optimal fractionation strategies across different lesion types. Adding to this complexity is the fact that patient and lesion characteristics can vary widely, such as in lesion thickness, vascularization, baseline oxygenation, or the kinetics of PpIX accumulation, and these factors may influence how a lesion responds to treatment. This variability suggests that tailoring parameters such as the dark interval duration, the initial light dose, or the total fluence could help maximize the sensitization phase without triggering premature and irreversible necrosis, particularly in lesions with low PS levels or poor perfusion. In this context, understanding how these variables impact treatment outcomes becomes especially relevant when considering more challenging lesions, such as deeper, thicker, or hypovascular tumors. Evidence from studies in nBCC and other difficult to treat conditions suggests that, while fractionation can still offer benefits, its effectiveness may be limited to patients with favorable lesion characteristics (e.g., small tumor diameter, superficial location). Thus, its generalizability outside of superficial dermatologic lesions warrants further investigation.

Another point of discussion involves the logistical implications of fractionated treatment. Extending session duration to accommodate dark intervals may reduce throughput in high-volume clinics, and not all treatment centers possess the technological infrastructure for timed, automated fractionated illumination. Additionally, although patient tolerability is often reported as improved, especially in terms of pain, this may vary depending on lesion location and individual sensitivity.

Finally, in a broader photobiological context, it is important to note that light exposure can also engage systemic neuroendocrine and immune networks. Recent advances in the emerging field of photo-neuro-immuno-endocrinology have highlighted how ultraviolet radiation regulates whole-body homeostasis, including the hypothalamic–pituitary–adrenal axis, neuroendocrine mediators, and immune modulation [58,59,60]. These findings expand our understanding of light as a biological regulator that acts not only at the tissue level, as in PDT, but also at the organismal level through integrated signaling pathways involving the brain, endocrine glands, and immune system. Although mechanistically distinct from PDT, such discoveries underscore the pleiotropic nature of light as a therapeutic agent. Taken together, they suggest that ongoing research in dermatologic PDT may ultimately intersect with broader photobiological paradigms, supporting the concept that light-based interventions can exert both local oncological control and systemic physiological benefits.

In conclusion, light fractionation has emerged as a biologically robust and clinically validated strategy to improve the performance of PDT. However, its full potential will only be realized through protocol standardization, mechanistic exploration, and broader clinical implementation. As PDT continues to evolve, light fractionation is likely to play an increasingly central role in refining its efficacy, selectivity, and patient-centered outcomes.

## Figures and Tables

**Figure 1 ijms-26-08054-f001:**
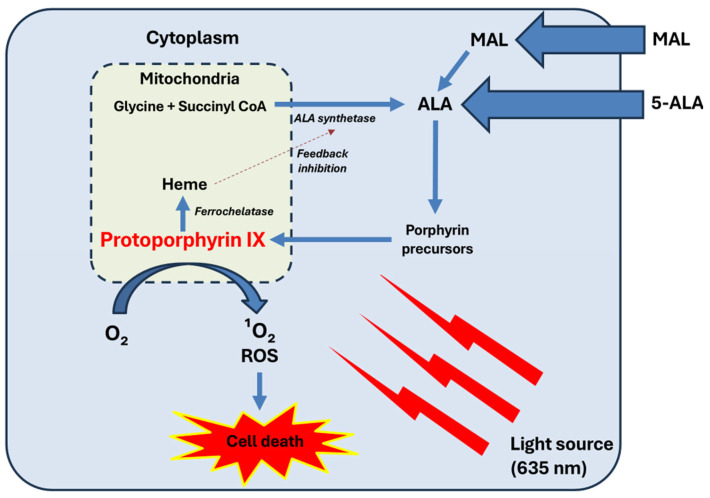
Mechanism of action of PDT. Exogenous ALA or MAL are taken up by cells. MAL needs to be hydrolyzed to ALA by intracellular esterases. ALA enters the heme biosynthesis pathway. Within the mitochondria, glycine and succinyl-CoA are converted into ALA by the enzyme ALA synthetase. This step is tightly regulated by negative feedback from heme, preventing excessive production of intermediates. ALA progresses through the porphyrin biosynthetic pathway, eventually forming PpIX. In normal conditions, PpIX is converted into heme by ferrochelatase. However, in tumor cells, reduced ferrochelatase activity leads to the preferential accumulation of PpIX. Upon exposure to a light source (usually at 635 nm), PpIX becomes activated, leading to the generation of singlet oxygen (^1^O_2_) and other ROS, such as superoxide anion radicals (O^2−^), hydrogen peroxide (H_2_O_2_), and hydroxyl radicals (•OH). These ROS induce oxidative damage to cellular structures, resulting in selective tumor cell death. Reproduced from E Garcia-Mouronte et al. [8].

**Figure 2 ijms-26-08054-f002:**
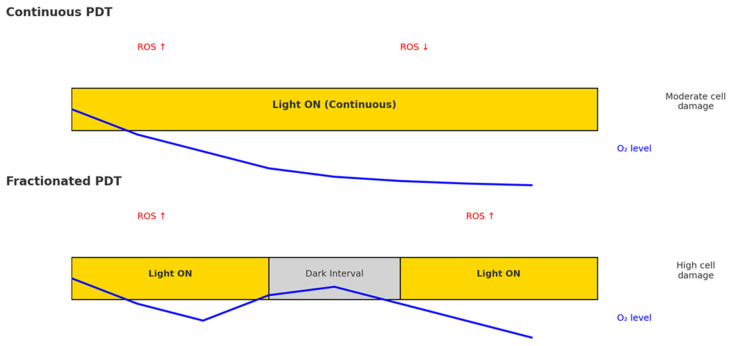
Schematic comparison between continuous and fractionated PDT. In continuous PDT, uninterrupted illumination causes rapid oxygen consumption, leading to progressive tissue hypoxia and reduced ROS production over time, which limits cytotoxic efficacy. In fractionated PDT, an initial light exposure is followed by a dark interval that permits partial tissue reoxygenation before a second illumination. Red arrows indicate the relative dynamics of ROS generation (increase at the start of illumination, decrease during sustained exposure, and renewed increase after reoxygenation in fractionated PDT). Blue line indicates tissue oxygen levels.

**Figure 3 ijms-26-08054-f003:**
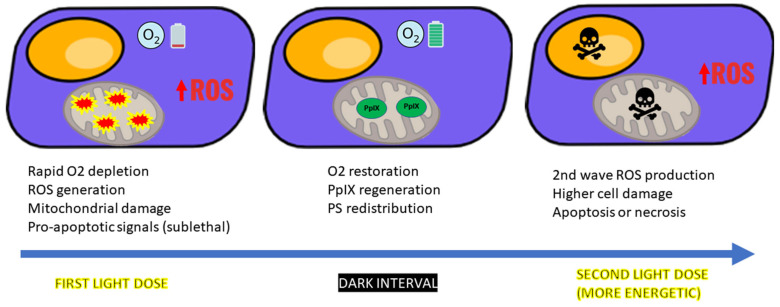
Sequential cellular events during light-fractionated PDT. The first light dose induces rapid oxygen depletion and subsequent ROS generation. This causes mitochondrial damage, including partial membrane permeabilization, depolarization, and the initial release of cytochrome c into the cytosol, triggering early activation of pro-apoptotic pathways (BAX translocation, partial caspase-3 activation). During the dark interval, tissue oxygenation is restored, protoporphyrin IX (PpIX) is synthesized through ongoing heme biosynthesis, and photosensitizer (PS) redistribution occurs. The subsequent, more energetic second light dose triggers a second wave of ROS production, resulting in greater mitochondrial and cellular damage, leading to apoptosis or necrosis. In the figure, the cell nucleus is shown in orange and the mitochondria in gray.

**Table 1 ijms-26-08054-t001:** Summary of clinical studies evaluating light fractionation in PDT across various dermatologic conditions. The table includes clinical indication, number of lesions, PS used, illumination protocols, comparators, and outcomes [13,15,16,17,18,32,33,34,35,36,37,38,39].

	Clinical Indication	Number of Lesions	Photo- Sensitizer	Fractionation Protocol	Comparator	Results
Sotiriou et al. [16]	AK	266	ALA	20 + 80 J/cm^2^, 2 h dark interval	75 J/cm^2^ × 2 (7 days apart)	Higher CR at 3 and 12 months (93.8% vs. 85.4%)
de Haas et al. [15]	AK, BD, sBCC, nBCC	552 (70 AK, 32 BD, 430 sBCC, 20 nBCC)	ALA	20 + 80 J/cm^2^, 2 h interval	None	CR: AK 98%, BD 84%, sBCC 97%, nBCC 80% at 2 years
Puizina-Ivić et al. [17]	AK, BD	51 (36 AK, 15 BD)	ALA	50 + 50 J/cm^2^, 2 h dark interval	100 J/cm^2^ single illumination	Significantly less residual tumor in fractionated group at 24 weeks (4% vs. 73%)
de Haas et al. [13]; C. de Vijlder et al. [32]	sBCC	505	ALA	20 + 80 J/cm^2^, 2 h dark interval	75 J/cm^2^ single illumination	CR: 97% vs. 89% at 12 months; 88% vs. 75% at 5 years
Kessels et al. [18]; van Delft et al. [33]	sBCC	162	ALA vs. MAL	ALA: 20 + 80 J/cm^2^	MAL: 37 J/cm^2^ × 2 (7 days apart)	Higher CR at 12 months (92.3% vs. 82.4%; not statistically significant), but lower long-term tumor-free survival after 5 years (70.7% vs. 76.5%)
Roozeboom et al. [34]; Mosterd et al. [35]	nBCC	173	ALA	75 + 75 J/cm^2^, 1 h dark interval; 3 weeks after debulking	Surgical excision	Higher recurrence at 5 years (30.7% vs. 2.3%). Better recurrence-free survival with PDT in tumors ≤ 0.7 mm (94.4% vs. 65%)
de Haas et al. [36]	BD	50	ALA	20 + 80 J/cm^2^, 2 h dark interval	75 J/cm^2^ single illumination	Higher CR at 12 months (88% vs. 80%; not statistically significant)
Suárez-Pérez et al. [37]	Actinic cheilitis	10	MAL	20 + 80 J/cm^2^, 2 h dark interval	None	CR: 80% at 3 months; 60% at 18 months.
Smits et al. [38]	Psoriasis (stable plaques)	8	ALA	2 + 8 J/cm^2^ weekly × 4	Placebo	Clinical + histological improvement; Koebnerization (25%)
Khan et al. [39]	Cutaneous leishmaniasis	104	MAL	30 + 30+ 30 J/cm^2^ (0 h, 2 h, 16 h)	90 J/cm^2^ single illumination	Higher CR at 9 months (91.4% vs. 76%); less pain

## Data Availability

No new data were created in this study. Data sharing is not applicable to this article.

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
