# Peer review of "Optimization of Photodynamic Therapy in Dermatology: The Role of Light Fractionation"

_ijms, 2025, doi:10.3390/ijms26168054_

Round 1
Reviewer 1 Report
Comments and Suggestions for Authors
1.Please increase the screening strategy for literature.
2.Please add references from the past 3 years. The references in this article are relatively old and the innovation of the research is slightly low.
3.Please add a mechanism pathway diagram。
4.What are the specific molecular and cellular mechanisms by which light fractional irradiation enhances the efficacy of photodynamic therapy (PDT)? The article mentions multiple possible mechanisms, such as increased generation of reactive oxygen species (ROS) by tissue reoxygenation and increased sensitivity of partially damaged cells to subsequent light exposure. However, the interactions between these mechanisms and their specific manifestations in different cell types and lesions still require further research and validation, and the current explanation in this article is relatively shallow.
5.Does the combination of light fractional irradiation and other treatment methods (such as lasers, drugs, etc.) have a synergistic effect in the treatment of skin tumors? The article mainly discusses the application of light fractional irradiation alone, but in practical clinical practice, the combination therapy strategy may further improve the treatment effect. Please further add relevant literature for comparison.
Author Response
We appreciate the reviewer’s careful evaluation of our work and the constructive suggestions provided. As part of this revision, and primarily, we have modified the title to “Optimization of Photodynamic Therapy in Dermatology: The Role of Light Fractionation”, which we believe more accurately reflects the scope and focus of the review.
Comment 1: Please increase the screening strategy for literature.
Response: Following your recommendation, we have substantially expanded our literature search to ensure a more comprehensive and updated coverage of the topic. This has led to the inclusion of multiple additional studies, which strengthen the scientific foundation of our review. As a result, the total number of references has increased from 29 in the original submission to 51 in the revised version. These new references have been integrated throughout the manuscript where relevant.
Comment 2: Please add references from the past 3 years. The references in this article are relatively old and the innovation of the research is slightly low.
Response: As mentioned in our response to Comment 1, a substantial part of the additional references incorporated in the revised manuscript are from recent years, thus improving the currency of the literature cited. Nevertheless, it should be noted that, despite an exhaustive review of the literature, there has unfortunately been limited progress in this specific area in the last years. Both molecular and clinical studies focusing on light‑fractionated PDT remain relatively scarce, and many of the most relevant and informative works on this topic date back further. We have nonetheless ensured that all recent contributions of scientific value have been included and appropriately integrated into the revised version.
Comment 3: Please add a mechanism pathway diagram
Response: We appreciate the reviewer’s suggestion. In the revised manuscript, we have included several new figures to better illustrate the mechanistic aspects of our review. One of these figures explains the fundamentals of photodynamic therapy (PDT), another depicts the concept of light fractionation, and an additional figure outlines the possible biological and molecular mechanisms through which light‑fractionated PDT may exert its effects. We believe these additions improve the clarity and visual impact of the manuscript.
Comment 4: What are the specific molecular and cellular mechanisms by which light fractional irradiation enhances the efficacy of photodynamic therapy (PDT)? The article mentions multiple possible mechanisms, such as increased generation of reactive oxygen species (ROS) by tissue reoxygenation and increased sensitivity of partially damaged cells to subsequent light exposure. However, the interactions between these mechanisms and their specific manifestations in different cell types and lesions still require further research and validation, and the current explanation in this article is relatively shallow.
Response: We thank the reviewer for this valuable observation. In the revised manuscript, we have aimed to include the studies that provide the most relevant insights into the molecular and cellular mechanisms underlying the effects of light‑fractionated PDT. To improve clarity and depth of explanation, we have reorganized the structure of the manuscript by merging the former Section 4 (Preclinical Evidence) and Section 6 (Biological and Molecular Mechanisms Underlying the Effects of Light Fractionation in PDT) into a single, more comprehensive section entitled “Mechanistic Insights from Preclinical Models of Light Fractionation in PDT.” This restructured section integrates the available preclinical experimental evidence with mechanistic explanations, as much of our current understanding of how light fractionation enhances PDT efficacy is derived from such studies. It covers the principal hypotheses, including ROS generation through tissue reoxygenation, increased sensitivity of sublethally damaged cells to subsequent illumination, PpIX redistribution vascular effects. However, we were unable to find sufficient experimental or clinical evidence in the literature to explain in detail how these mechanisms may interact or how they might vary between different cell types and lesion contexts. We have addressed this point at the end of the first paragraph of the Discussion.
Comment 5: Does the combination of light fractional irradiation and other treatment methods (such as lasers, drugs, etc.) have a synergistic effect in the treatment of skin tumors? The article mainly discusses the application of light fractional irradiation alone, but in practical clinical practice, the combination therapy strategy may further improve the treatment effect. Please further add relevant literature for comparison.
Response: We appreciate the reviewer’s suggestion. In the revised manuscript, we have substantially expanded the Future Directions section to address this point. This now includes a discussion of complementary treatments that have already been evaluated in combination with light‑fractionated PDT, as well as those which, although not yet tested in this specific context, could represent promising directions for future research if combined with light‑fractionated PDT. Relevant literature has been incorporated to support these additions.
Reviewer 2 Report
Comments and Suggestions for Authors
Light Fractionation Enhances the Efficacy of Photodynamic Therapy in Dermatologic Oncology is a review based on selection of current papers.
Section Fundamentals of PDT is missed some figures and reaction schemes.
Section Light fractionation concept can use some figure too.
Section Preclinical evidence supporting light fractionation in PDT please add table and more references.
Section Biological and Molecular Mechanisms Underlying the Effects of Light Fractionation in PDT needs to be removed or properly described. Can not be 5 sentences about this.
In my opinion too many sections without figures and deep analysis.
Sincerely
Author Response
We appreciate the reviewer’s careful evaluation of our work and the constructive suggestions provided. As part of this revision, and primarily, we have modified the title to “Optimization of Photodynamic Therapy in Dermatology: The Role of Light Fractionation”, which we believe more accurately reflects the scope and focus of the review.
Comment 1: Section Fundamentals of PDT is missed some figures and reaction schemes.
Response: Following your suggestion, we have added Figure 1 in the Fundamentals of PDT section. This figure illustrates the key components of photodynamic therapy, the photochemical activation of the photosensitizer, and the subsequent generation of reactive oxygen species, providing a clear visual summary of the process.
Comment 2: Section Light fractionation concept can use some figure too.
Response: We thank the reviewer for this suggestion. In the revised manuscript, we have added Figure 2 in the Light Fractionation Concept section. This figure visually explains the principles of light fractionation, illustrating the delivery of illumination in multiple fractions separated by a dark interval, the associated tissue reoxygenation, and the enhanced therapeutic effect compared to continuous illumination.
Comment 3: Section Preclinical evidence supporting light fractionation in PDT please add table and more references.
Response: We thank the reviewer for this suggestion. In the revised manuscript, we have merged the previous Sections 4 (Preclinical Evidence) and 6 (Biological and Molecular Mechanisms Underlying the Effects of Light Fractionation in PDT) into a single, more comprehensive section entitled “Mechanistic Insights from Preclinical Models of Light Fractionation in PDT”. We have also incorporated additional, relevant literature into this section, with further related references included in the Future Directions section. However, we do not consider that a summary table would be appropriate here, as the available studies are highly heterogeneous in design, endpoints, and outcome measures. In our view, such heterogeneity would limit comparability and might obscure rather than clarify the underlying concepts for the reader.
Comment 4: Section Biological and Molecular Mechanisms Underlying the Effects of Light Fractionation in PDT needs to be removed or properly described. Can not be 5 sentences about this.
Response: We thank the reviewer for this observation. As mentioned above, in the revised manuscript we have merged the previous Sections 4 and 6 into a single section. This new section now provides a detailed description of the available preclinical evidence together with the proposed biological and molecular mechanisms. In addition, we have included Figure 3, which offers a visual summary of the biological mechanisms through which light‑fractionated PDT may produce its effects.
Comment 5: In my opinion too many sections without figures and deep analysis.
Response: In this revised version, we have enriched the manuscript with additional figures and expanded analyses across several sections. New illustrations now explain the fundamentals of PDT (Figure 1), the concept of light fractionation (Figure 2), and the proposed biological and molecular mechanisms involved (Figure 3). The discussion in key areas has also been expanded to provide greater analytical depth. We hope these additions address the reviewer’s concerns and enhance the clarity and overall quality of the work.
Round 2
Reviewer 1 Report
Comments and Suggestions for Authors
I am satisfied with the author's reply and would like to express my gratitude
Author Response
We sincerely thank you for your positive feedback and valuable input.
Reviewer 2 Report
Comments and Suggestions for Authors
Thank you
Author Response

(The authors gave the same response as above.)
